# Abundance of Microvascular Endothelial Cells Is Associated with Response to Chemotherapy and Prognosis in Colorectal Cancer

**DOI:** 10.3390/cancers13061477

**Published:** 2021-03-23

**Authors:** Masanori Oshi, Michelle R. Huyser, Lan Le, Yoshihisa Tokumaru, Li Yan, Ryusei Matsuyama, Itaru Endo, Kazuaki Takabe

**Affiliations:** 1Roswell Park Comprehensive Cancer Center, Department of Surgical Oncology, Buffalo, NY 14263, USA; masanori.oshi@roswellpark.org (M.O.); michelle.huyser@roswellpark.org (M.R.H.); Lan.Le@RoswellPark.org (L.L.); Yoshihisa.Tokumaru@roswellpark.org (Y.T.); 2Department of Gastroenterological Surgery, Yokohama City University Graduate School of Medicine, Yokohama 236-0004, Japan; ryusei@yokohama-cu.ac.jp (R.M.); endoit@yokohama-cu.ac.jp (I.E.); 3Department of Surgery, Jacobs School of Medicine and Biomedical Sciences, State University of New York, Buffalo, NY 14263, USA; 4Department of Surgical Oncology, Graduate School of Medicine, Gifu University, 1-1 Yanagido, Gifu 501-1194, Japan; 5Roswell Park Comprehensive Cancer Center, Department of Biostatistics & Bioinformatics, Buffalo, NY 14263, USA; li.yan@roswellpark.org; 6Division of Digestive and General Surgery, Niigata University Graduate School of Medical and Dental Sciences, Niigata 951-8520, Japan; 7Department of Breast Surgery, Fukushima Medical University School of Medicine, Fukushima 960-1295, Japan; 8Department of Breast Surgery and Oncology, Tokyo Medical University, Tokyo 160-8402, Japan

**Keywords:** biomarker, colorectal cancer, GSEA, tumor microenvironment, survival, transcriptome, xCell

## Abstract

**Simple Summary:**

The generation of pathologic, immature, and dysfunctional vessels by angiogenesis is a well-known mechanism of metastasis and has been a therapeutic target for colorectal cancer (CRC). In this study, we investigated the clinical relevance of microvascular endothelial (mvE) cells in CRC by analyzing tumor gene expression profiles of large patient cohorts. We found that the abundance of mvE cells does not mirror angiogenesis, but rather is associated with the number of mature blood vessels in the tumor microenvironment and predicts the response to chemotherapy as well as patient survival in CRC. This is the first study suggesting the clinical relevance of mvE cells in CRC.

**Abstract:**

The generation of pathologic, immature, and dysfunctional vessels by angiogenesis is a mechanism of metastasis that has been a therapeutic target for colorectal cancer (CRC). In this study, we investigated the clinical relevance of intra-tumoral microvascular endothelial (mvE) cells in CRC using the xCell algorithm on transcriptome. A total of 1244 CRC patients in discovery and validation cohorts were analyzed. We found that an abundance of mvE cells did not mirror angiogenesis but reflected mature blood vessels because it was significantly associated with a high expression of vascular stability-related genes, including sphingosine-1-phosphate receptor genes and pericytes. Epithelial–mesenchymal transition and myogenesis gene sets were enriched in mvE cell abundant CRC, while mvE cell-less CRC enriched cell proliferation, oxidative phosphorylation, and protein secretion gene sets. mvE cell abundant CRC was associated with infiltration of M2 macrophages, dendritic cells, and less gamma-delta T cells (all *p* < 0.001), but not with the interferon-γ response. mvE cell abundant CRC was significantly associated with worse patient survival in CRC. Interestingly, mvE cell abundant CRC was significantly associated with a high response rate to chemotherapy (*p* = 0.012) and worse patient survival for those that did not receive chemotherapy. However, there was no survival difference in patients who underwent chemotherapy. In conclusion, we estimated the abundance of mvE cells using the xCell algorithm on tumor transcriptome finding its association with the number of mature blood vessels in a tumor microenvironment and its ability to predict response to chemotherapy, thereby patient survival in CRC.

## 1. Introduction

Colorectal cancer (CRC) is diagnosed in more than 1.2 million patients worldwide every year, which is approximately 10% of all types of cancer [1]. More than one-fourth of those patients die within five years of diagnosis, mainly due to systemic metastases [2,3]. Multiple mechanisms have been identified during the process of metastasis [4]. Angiogenesis, the development of numerous new blood vessels around and in the tumor, is one of the mechanisms of metastasis that contributes to cancer cell dissemination and has been a therapeutic target for CRC. Bevacizumab, a vascular endothelial growth factor (VEGF) antagonist, commercially known as Avastin, suppresses angiogenesis thereby prolonging survival of metastatic CRC patients when combined with chemotherapy [5]. Additional newer drugs also target angiogenesis, which include aflibercept, a vascular endothelial growth factor (VEGF) and placental growth factor (PIGF)-antagonist, and regorafenib, a multi-tyrosine kinase inhibitor, and have been shown to significantly improve the progression-free survival of CRC in Phase III randomized trials [6,7,8]. 

Angiogenesis produces pathologic blood vessels that are immature and not wrapped with pericytes, so they do not function as mature blood vessels with efficient delivery of oxygen and drugs. The gold standard to quantify angiogenesis is by the microvessel density of blood vessels using immunohistochemistry or immunofluorescence [9]. Our group previously reported that sphingosine kinase 1 (SphK1)-produced sphingosine-1-phosphate (S1P), a pleiotropic bioactive lipid mediator, plays a critical role in breast-cancer-induced angiogenesis using microvessel density [10]. S1P produced by SphK1 is exported out of cancer cells [11,12] and binds to S1P receptors 1 (S1PR1) and 3 (S1PR3) on blood vessels to promote angiogenesis [13,14]. Morphological methodologies such as microvessel density have the advantage of gaining information regarding the structure of the cancer tumor; however, they are often laborious, time-consuming, and their interpretation is pathologist-dependent, leading to varying results. During the same study, we showed that quantification of vascular endothelial cells by flow cytometry correlated very well with microvessel density [10]; however, flow cytometry requires fresh samples, which are labor- and cost-intensive, thus unsuitable for analyzing an immense number of samples.

In recent years, there have been remarkable advances in the understanding of the tumor microenvironment (TME) using transcriptome and improved gene sequence technologies [15,16]. The transcriptome of a bulk tumor allows simultaneous analysis of multiple cell markers and cell functions as well as interactions between cancer cells and the TME. Analyses of a single gene expression are often difficult to interpret due to complex relationships between multiple cells in the TME. To overcome this challenge, several approaches have been developed using different algorithms. Gene set enrichment analysis (GSEA) has been used to estimate the activated pathways of cancer cells and stromal cells in the TME using multiple related genes [17,18]. The xCell algorithm allows estimation of the relative proportions of 64 types of cells, such as immune cells and stromal cells in the TME, using the transcriptome of a bulk tumor [19]. We previously reported the clinical relevance of multiple types of cells in the TME using the xCell algorithm, including plasmacytoid dendritic cells, which correlate with better survival [20], and the association of regulatory T cells with the response to neoadjuvant chemotherapy in triple-negative breast cancer [21]. Cancer-associated fibroblasts were associated with curative resection in pancreatic ductal adenocarcinoma [22]. The xCell algorithm allows quantification of microvascular endothelial (mvE) cells, allowing us to estimate the amount of angiogenesis occurring in a given TME with transcriptome and without laborious pathological analyses and costly flow cytometry.

In this study, we hypothesized that the abundance of mvE cells estimated by xCell analyses on tumor transcriptome is associated with the number of mature blood vessels in the TME and will predict responses to chemotherapy and patient survival in CRC.

## 2. Results

### 2.1. Abundance of Microvessel Endothelial (mvE) Cells Is Associated with Elevated Expressions of Blood-Vessel-Related Genes, Such as Vascular Endothelial Growth Factor (VEGF), Endothelial Cell, and Vascular Stability, in Colorectal Cancer 

The gold standard to quantify angiogenesis is by microvessel density in a peritumoral area. We previously reported that quantification of vascular endothelial cells by flow cytometry was as accurate as microvessel density [10]. In this study, we investigated whether the amounts of microvascular endothelial (mvE) cells estimated by the xCell algorithm (Appendix A) adequately capture the features of vasculogenesis in tumor microenvironments (TMEs) by comparing mvE scores and expressions of several angiogenesis-related genes in the Cancer Genome Atlas (TCGA) and GSE39582 cohorts (Appendix A). We investigated vascular endothelial growth factor (VEGF)-related genes; VEGF-A, VEGF-B, and VEGF-C, and endothelial cell marker-related genes; CD31/PECAM1, von-Willebrand factor (vWF), and vascular stability-related genes; TKE1, TIE2/TEK, VE-Cadherin/CDH5, and Claudin 5/CLDN5, where the choice of genes was based on our previous reports [23,24]. High mvE CRC was significantly associated with high expression of all the genes, except for VEGF-A (Figure 1A–C). Interestingly, high mvE CRC was significantly associated with an abundance of pericytes consistently in both cohorts (Figure 1D; both *p* < 0.001). Together, our results suggest that high mvE CRC is associated not only with angiogenesis that develops immature vessels, but also mature blood vessels with high expression of vascular-stability-related genes and pericytes.

### 2.2. mvE Cell Abundant CRC Was Significantly Associated with High Fraction of Stroma Score and Expression of Sphingosine-1-Phosphate (S1P)-Related Genes

Next, we investigated the association of the mvE cells with the other stromal cells in the TME. mvE cell abundant CRC was significantly associated with a high stroma score in both cohorts (Figure 2A). mvE cells were associated with less fraction of epithelial cells and not of fibroblasts (Appendix A). These results suggested that mvE cells may affect the number of stromal cells in the TME. Our group and others previously reported that S1P produced by SphK1 plays a critical role in angiogenesis [13,14]. As shown in Figure 2B, mvE cell abundant CRC was significantly associated with high expression of several S1P-related genes. In particular, sphingosine-1-phosphate kinase receptor 1 (S1PR1), and 3 (S1PR3), both known to be expressed in mature blood vessels, were consistently associated with mvE cell abundant CRC in both cohorts. 

### 2.3. mvE Cell Abundant CRC Enriched Epithelial–Mesenchymal Transition (EMT) and Myogenesis Gene Sets, and Low mvE CRC Enriched Cell Proliferation-Related Gene Sets, but Did Not Correlate with Angiogenesis 

Since mvE cells are associated with expressions of the angiogenesis-related genes of CRC, we expected the number of mvE cells to be associated with cancer-promoting signaling pathways, including angiogenesis. To this end, gene set enrichment analysis (GSEA) was performed to explore the associated gene sets enriched in low or high mvE cell CRC with hallmark gene sets. GSEA showed that high mvE CRC enriched epithelial–mesenchymal transition (EMT) and myogenesis gene sets consistently in both TCGA and GSE39582 cohorts (Figure 3A). Surprisingly, an abundance of mvE cells did not correlate with angiogenesis (Figure 3B; Spearman’s rank correlation test (*r*) = 0.245 and 0.260, respectively, both *p* < 0.01). However, low mvE cell CRC consistently enriched E2F target, MYC target v1, oxidative phosphorylation, and protein secretion gene sets in both cohorts (Figure 3C). These results implied that the abundance of mvE cells is not a mere reflection of angiogenesis but is a marker of a unique type of CRC that is less proliferative.

### 2.4. Abundance of mvE Cells Was Associated with High Fraction of Dendritic Cells (DCs) and M2 Macrophages

Since an abundance of mature blood vessels promotes immune cell infiltration [23,24], we hypothesized that the abundance of mvE cells is associated with the fraction and function of immune cells. The xCell algorithm was used to investigate the relationship between mvE cells and immune cell infiltrations in the TME. We found that mvE cells were significantly associated with a high fraction of dendritic cells (DC) and M2 macrophages, and a low fraction of γδT in both TCGA and GSE39582 cohorts (Figure 4A). High mvE cells were significantly associated with a high fraction of CD8^+^ T cells and a low fraction of CD4^+^ T cells in the GSE39582, but not in the TCGA cohort. M1 macrophages showed a significant difference between low and high mvE cell groups in the TCGA but not in the GSE39582. High mvE was also associated with high lymphocyte infiltration, leukocyte fraction, T cell receptor (TCR), and B cell receptor (BCR), which are the scores calculated by Thorsson et al. in the TCGA cohort (Appendix A). However, there was no association between mvE cells and interferon (IFN)-γ response score, which represents immune activity. GSEA showed that immune-related gene sets; and inflammatory response, IFN-α, and IFN-γ response did not enrich in low or high mvE tumor groups in either cohort (Figure 4B). These findings suggested that the abundance of mvE cells in the TME is not associated with unidirectional anticancer immunity.

### 2.5. mvE Cell Abundant CRC Was Significantly Associated with Worse Survival 

As the TME contributes to cancer growth, cancer progression, drug resistance, and metastasis, we next investigated the association of the number of mvE cells with clinical aggressiveness in CRC. We found that mvE cells were not associated with the American Joint Committee on Cancer (AJCC) stage (Figure 5A), tumor size (T-category), or lymph node metastasis (N-category) in either TCGA or GSE39582 cohorts (Appendix A). Although mvE cells were not associated with genomic status (microsatellite stable (MSS), microsatellite instability (MSI)-low, and MSI-high) (Appendix A) and mutation of *BRAF*, *KRAS*, and *NRAS* genes (Appendix A), mvE cells in mucinous adenocarcinoma were significantly higher than in adenocarcinoma (Appendix A). The association between mvE cells and expression of vessel- and S1P-related genes and gene sets in each mucinous adenocarcinoma and adenocarcinoma cohorts (Appendix A) demonstrated mvE cell abundant CRC was significantly associated with worse overall survival (OS), disease-specific survival (DSS), and progression-free survival (PFS) in the TCGA cohort (Figure 5B; *p* = 0.038, 0.001, and 0.002, respectively). The results for OS and PFS were validated independently by another CRC cohort, GSE39582 (Figure 5B; *p* = 0.002 and 0.019, respectively). Furthermore, the prognostic value of mvE cells was independent of other factors, including age (more than 65 years old or not, subtype (mucinous adenocarcinoma or adenocarcinoma), genomic status (MSI or MSS), and AJCC stage (III/IV or I/II)) of the DSS in the TCGA cohort (Appendix A). These findings suggested that the abundance of mvE cells is significantly associated with worse patient survival in CRC. 

### 2.6. mvE Cell Abundant CRC Was Significantly Associated with Better Response to Chemotherapy, and There Was No Survival Difference with mvE Cell Amount When Treated by Chemotherapy 

Angiogenesis in tumors is thought to generate pathologic and dysfunctional blood vessels that do not contribute to circulation or drug delivery to the TME [25]. As mvE cell abundant CRC is associated with increased pericyte, and thus mature blood vessels, we hypothesized that mvE cell abundant CRC is rich with mature blood vessels that could contribute to drug delivery. To test our hypothesis, we investigated the relationship between the abundance of mvE cells and their response to chemotherapy in CRC. We found that mvE cell abundant CRC was significantly associated with a better response rate to chemotherapy in the GSE28072 cohort (Appendix A), which demonstrated patient response after FOLFOX (Figure 6A; *p* = 0.012). mvE cell abundant metastatic tumors show similar trends as the primary tumor but showed no statistically significant difference (*p* = 0.391). Next, we compared survival outcomes between low and high mvE cell CRC in the cohort that did not undergo chemotherapy in the GSE39582 cohort. Interestingly, patients with mvE cell abundant CRC were significantly associated with worse survival in the cohort that did not receive chemotherapy (Figure 6B; OS: *p* < 0.001 and recurrence-free survival (RFS): *p* < 0.001), whereas there was no difference in the cohort that was treated with chemotherapy (Figure 6B; OS: *p* = 0.774 and RFS: *p* = 0.344). Together, these findings imply that mvE cell abundant CRC better responds to chemotherapy, although they are more biologically aggressive with worse survival.

## 3. Discussion

In this study, we investigated the clinical relevance of the abundance of microvascular endothelial (mvE) cells in CRC using the xCell algorithm on transcriptome of bulk tumors. mvE cell abundant CRC was significantly associated with high expression of multiple vessel-related genes including VEGF and endothelial cell marker-genes. We found that mvE cells were significantly associated with an abundance of pericytes and stromal cells in the TME, as well as vascular-stability- and S1P-related genes. This implies that mvE cells are not associated with pathological immature angiogenesis-derived vessels but with mature functional blood vessels. mvE cell abundant CRC enriched EMT and myogenesis gene sets; and mvE cell-less CRC enriched E2F target, MYC target v1, oxidative phosphorylation, and protein secretion gene sets. mvE cell abundant CRC was significantly associated with a high fraction of M2 macrophages and dendritic cells and a low fraction of γδT cells, but not with unidirectional anti-cancer immunity. mvE cell abundant CRC was significantly associated with worse patient survival. Finally, mvE cell abundant CRC was significantly associated with a better response to chemotherapy. Interestingly, there was no difference in survival regarding the number of mvE cells in patients who underwent chemotherapy. However, mvE cell abundant CRC was associated with worse survival in the non-treatment group, which agrees with a better chemotherapy response in the mvE cell abundant group. 

Angiogenesis is a known mechanism of cancer progression, which is a therapeutic target for CRC. Measurements of microvessel density using immunohistochemistry or immunofluorescence are the optimum standard to quantify angiogenesis [9]. Our group previously reported that flow cytometry of blood endothelial cells is as good as microvessel density in an animal experimental setting [10]. These methodologies are well-established and reliable; however, they often require a pathologist to interpret them, so they can be laborious, time-consuming, and costly. The results may significantly vary depending on the tumor site and by the pathologist’s subjective interpretation. Analysis using one slice section of a tumor does not necessarily reflect the angiogenesis of the entire tumor. Quantification by gene expression is another option; however, single gene expression may or may not represent the whole picture of angiogenesis in a tumor. In this study, we used the xCell algorithm to calculate the abundance of intra-tumoral mvE cells using the transcriptome of a bulk tumor. We previously reported the clinical relevance of several cells in the TME, such as CD8^+^, regulatory T cells, dendritic cells, and fibroblasts using the xCell algorithm [20,21,22,26].

It was thought that angiogenesis-derived vessels improve microcirculation in the TME and facilitate drug delivery to cancer cells, thus may be an indicator of treatment efficacy [27]. Now, angiogenesis-derived vessels are pathologic, immature, and dysfunctional and do not improve microcirculation nor drug delivery [25]. In this study, we found that the abundance of mvE cells did not enrich angiogenesis, meaning it does not mirror angiogenesis. However, an abundance of mvE cells was associated with pericytes and expression of vascular-stability-related genes, better drug response, and patient survival. The presence of intratumoral mvE indicates that mature functional blood vessels can contribute to drug supply within the TME. We speculate that an abundance of mvE cells may become a candidate as a predictive biomarker of chemotherapy response in CRC.

This study is not free of limitations. This is a retrospective study in design, although we used multiple patient cohorts to overcome bias. We used publicly available cohorts of previous studies; thus, our analyses were limited to the parameters gathered by the previous authors. The cohorts we examined vary by patient background and clinical characteristics, and there is also limited information on the classification of CRC, such as morphological aspects and subtypes. Our analyses were limited to measurements of gene expressions, whereas the gold standard to evaluate angiogenesis in the TME is microvessel density, which may also contribute to a difference in results. Furthermore, we were unable to investigate the association of the abundance of mvE cells and the effects of other treatments due to the lack of such data in the cohorts. Finally, we did not assess the biological mechanisms that underlie our findings.

## 4. Materials and Methods 

### 4.1. mRNAs Expression and Clinical Datao of Colorectal Cancer Cohorts

mRNA expression data and clinical features of patients with colorectal cancer were obtained from TCGA colorectal cancer (TCGA_COAD/READ; *n* = 595) cohort through the Genomic Data Commons Data Portal (GDC) (https://portal.gdc.cancer.gov, accessed on 3 November, 2019) [28]. Survival data were obtained from pan-Cancer Clinical Data Resource [29]. Clinical and mRNA expression data of CRC patients studied by Marisa et al. (GSE39582; *n* = 566) [30] were obtained from Gene Expression Omnibus (GEO) repository (http://www.ncbi.nlm.ih.gov/geo/, accessed on 5 November, 2020) as a validation cohort. For drug response analysis, data from Tsuji et al. (GSE28702; *n* = 83, regimen; FOLFOX) [31] were also obtained from the GEO repository. mRNA expression data were extracted from surgical specimens in all cohorts. The average gene value, with multiple probes and gene expression data, was transformed for log_2_ in all analyses.

### 4.2. Estimate the Fraction of Infiltrating Cells in Tumo Microenvironment

xCell score, which allowed us to estimate 64 types of cells in the TME using mRNA expression data [19], was calculated within cohorts using R (version 4.0.1, R Project for Statistical Computing). 

### 4.3. Gene Set Expression Analyses

As defined by the gene set enrichment analysis (GSEA) software (Lava version 4.0), false discovery rate (FDR) [32], gene sets with a false positive rate (FDR) of less than 25% were considered statistically significant. 

### 4.4. Statistical Analysis

All data analysis used R software (version 4.0.1). The top one-fourth was used as a cut-off to categorize patients into two groups based on mvE cells score within each cohort. *p*-value was calculated by Mann–Whitney U test or Fisher’s exact tests for group comparison analysis, and by log-rank test for survival analysis, as we described in each figure legend. *p*-value < 0.05 was taken statistically significant in all analyses. The Tukey-type boxplots show median and interquartile level values. 

## 5. Conclusions

We estimated the abundance of microvascular endothelial (mvE) cells using the xCell algorithm on tumor transcriptome and found that it is associated with the number of mature blood vessels in the tumor microenvironment (TME), and will predict response to chemotherapy and patient survival in colorectal cancer (CRC).

## Figures and Tables

**Figure 1 cancers-13-01477-f001:**
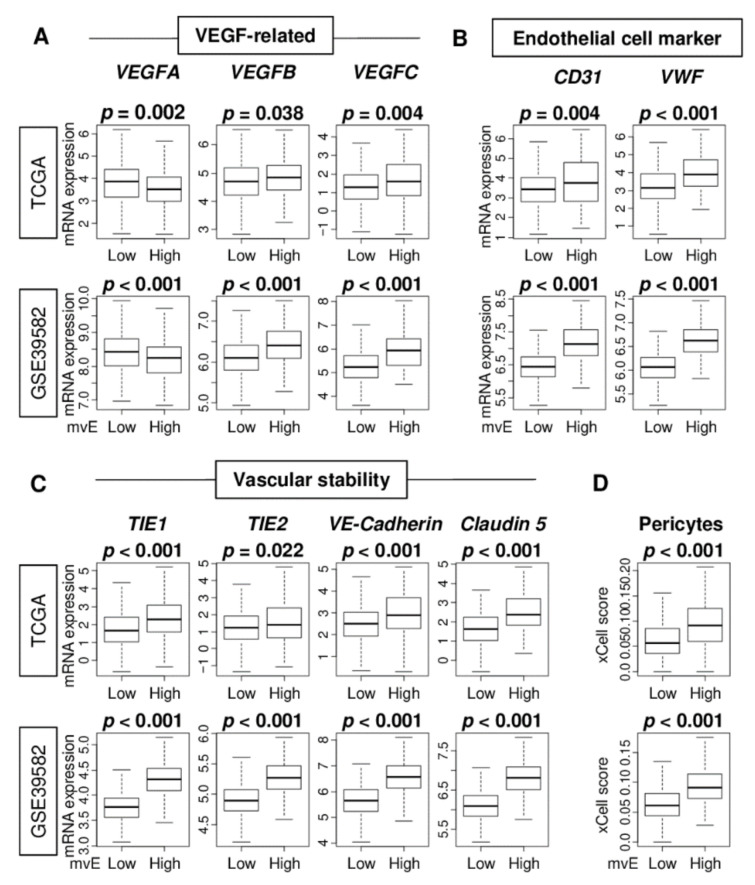
Association of the microvessel endothelial cells (mvE) with expression of vessel-related genes in the Cancer Genome Atlas (TCGA) and GSE39582 cohorts. Boxplots of the comparison of (**A**) vascular endothelial growth factor (VEGF)-related genes, VEGF-A, VEGF-B, and VEGF-C; and (**B**) endothelial cell-related genes, CD31 and VWF; and (**C**) vascular stability-related genes, TIE1, TIE2, VE-Cadherin, and Claudin 5; and (**D**) abundance of pericytes between high and low mvE groups. The top one-fourth was used as a cut-off to divide low and high groups for each cohort. *p*-value was calculated by the Mann–Whitney U test.

**Figure 2 cancers-13-01477-f002:**
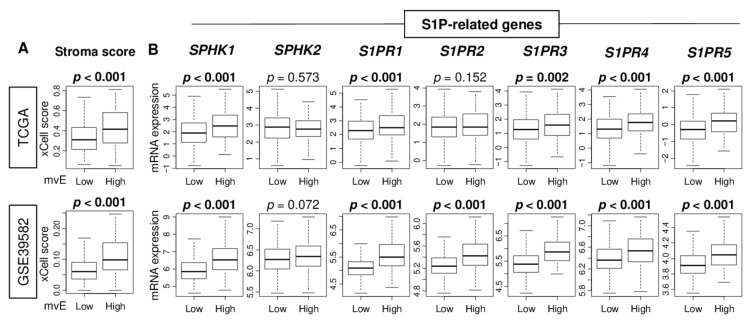
The association of the number of mvE cells with stromal cells and sphingosine-1-phosphate (S1P)-related genes. Boxplots of (**A**) the stromal score and (**B**) expression of S1P-related genes; SPHK1, SPHK2, S1PR1, S1PR2, S1PR3, S1PR4, and S1PR5, by high and low mvE cell groups in the TCGA and GSE39582 cohorts. The top one-fourth was used as a cut-off to divide low and high groups for each cohort. *p*-value was calculated by the Mann–Whitney U test. S1PR, Sphingosine-1-phosphate kinase receptor.

**Figure 3 cancers-13-01477-f003:**
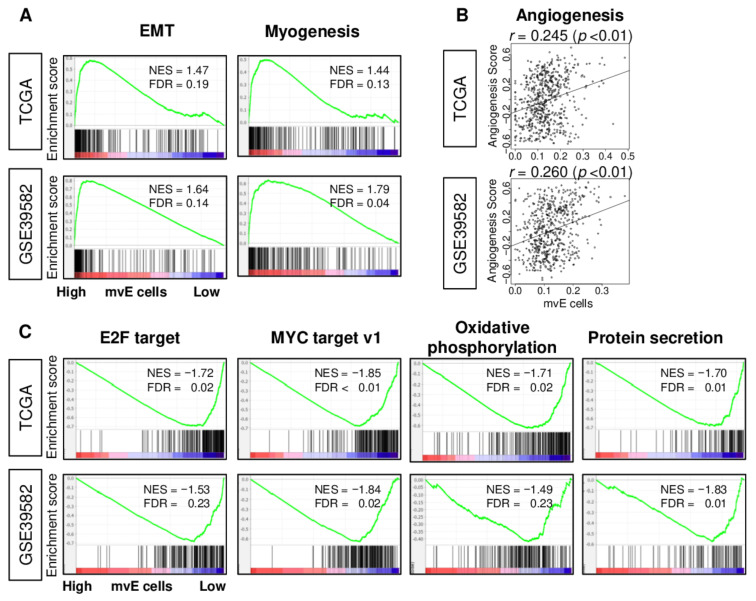
Gene set enrichment analysis (GSEA) of mvE cell CRC in the TCGA and GSE39582 cohorts. (**A**) Enrichment plots of gene sets enriched in high mvE cell CRC and (**B**) correlation plots between mvE cell score and angiogenesis score. (**C**) Enrichment plots of gene sets enriched in low mvE cell CRC. The top one-fourth was used as a cut-off to divide low and high groups for each cohort in (**A**) and (**C**). Significantly enriched gene sets were selected based on false discovery rate (FDR) *q*-value < 0.25. Spearman’s rank correlation test was used to analyze in (**B**). EMT; epithelial–mesenchymal transition, NES; normalized enrichment score.

**Figure 4 cancers-13-01477-f004:**
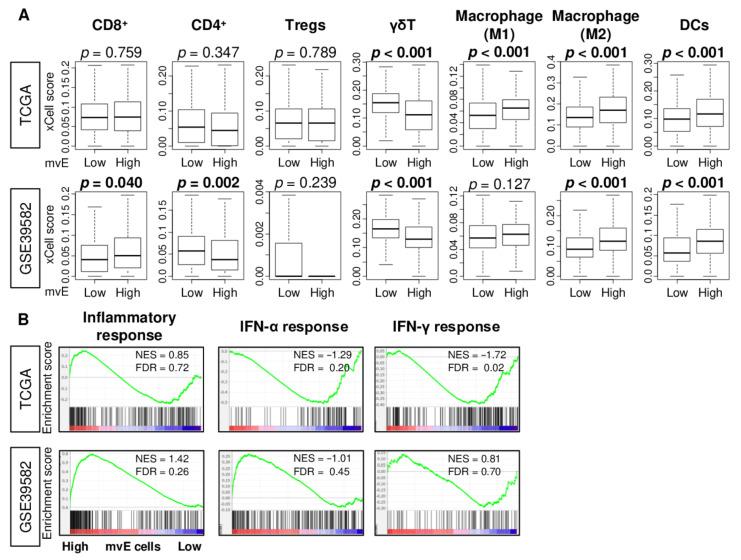
Association between amount of mvE cells and immunity of TME in the TCGA and GSE39582 cohorts. (**A**) Boxplots of the fraction of immune cells; CD8^+^ T cells, CD4^+^ T cells, M1 and M2 macrophages, dendritic cells (DC), γδT cells, and regulatory T cells (Tregs) by high and low mvE cell tumor groups in both cohorts. *p*-value was calculated by the Mann–Whitney U test. (**B**) Enrichment plots of immune-related gene sets; inflammatory response and interferon (IFN)-α and -γ response in both cohorts. The top one-fourth was used as a cut-off to divide low and high groups for each cohort. NES; normalized enrichment score; FDR, false discovery rate.

**Figure 5 cancers-13-01477-f005:**
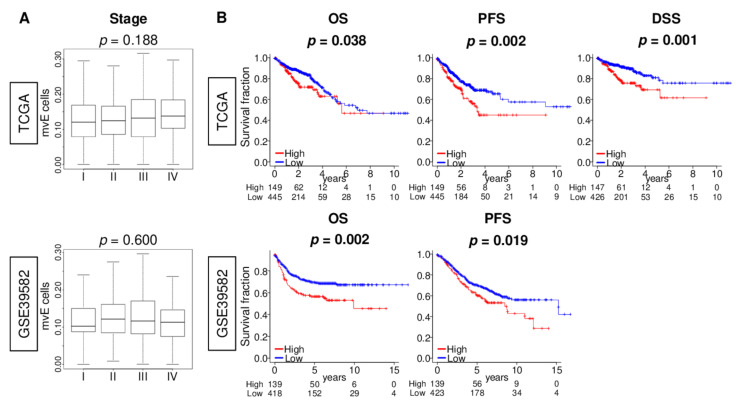
Association between amount of the mvE cells with clinical aggressiveness in the TCGA and GSE39582 cohorts. (**A**) Box plots of the mvE cells by the American Joint Committee on Cancer (AJCC) stage in both cohorts. *p*-value was calculated by Kruskal–Wallis test. (**B**) Kaplan–Meier curve plots of overall survival (OS) and progression-free survival (PFS) in both cohorts, and disease-specific survival (DSS) in the TCGA cohort by high (red) and low (blue) mvE cells groups with the *p*-value. *p*-value was calculated by the log-rank test. The top one-fourth was used as a cut-off to divide low and high groups for each cohort.

**Figure 6 cancers-13-01477-f006:**
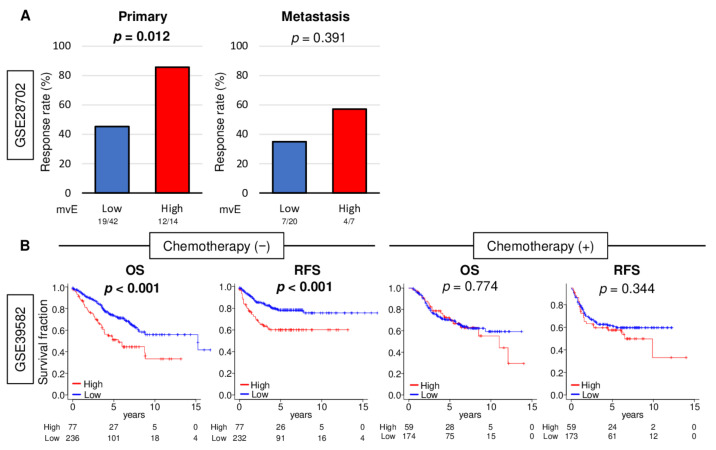
Association between amount of mvE cells with treatment response. (**A**) Bar plots of pathological response rate for chemotherapy for primary and metastasis between high (red) and low (blue) mvE cells score in the GSE28072 cohort. (**B**) Kaplan–Meier with *p*-value between high (red) and low (blue) mvE cells with overall survival (OS) and recurrence-free survival (RFS) in the treatment group and non-treatment group in the GSE39582 cohort. *p*-value was calculated by log-rank test.

## Data Availability

All data were from previous studies.

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
