# Peer review of "Abundance of Microvascular Endothelial Cells Is Associated with Response to Chemotherapy and Prognosis in Colorectal Cancer"

_cancers, 2021, doi:10.3390/cancers13061477_

Round 1
Reviewer 1 Report
In the presented work, the role of vascularization of CRC and it´s
environment are described and microvascular endothelial cells with association with response to chemotherapy and prognosis in colorectal Cancer is highlighted.
The text is well written and English is appropriate. Methods and figures are well done.
Some correction of spelling and grammar could be improved. In general, this work represents a new insight to the role of EGFR, endothelial cells and tumor environment.
I am just concerned about two aspects: Are there any molecular data like MSI/MSS or morphological subtypes that were different for vascularization of analysed CRC?
Regarding subtypes: Please check the independence of analysed markers or combinations in your cohort, as some subtypes of CRC are well known and show complete different reaction to therapy, in some cases regarding vascularization.
Also following the morphological aspects there are pictures /figures missing completely showing the difference in the growth and tumor pattern in the CRC cohort a swell as the visualization of tumor vascularization and differences. Pleas provide pictures of different aspects of your work on CRC.
Tasks:
- Improve and check spelling and grammar (there are no big issues).
- Improve and fulfil comments regarding figures (see above).
- Please clarify aspects of subtypes (independence of the analysed markers in serrated, mucinous, dedifferentiated, medullary or signet ring cell type of CRC.
- Please fulfil aspects regarding molecular data like MSI/MSS. In addition, BRAF and KRAS, NRAS aspects are missing.
- Some figures maybe also figures of morphological subtypes can be submit as supplement (editing required).
Reviewer 2 Report
Thank you for providing me with the opportunity to review this manuscript. Herein, the authors attempted to investigate the clinical relevance of microvascular endothelial (mvE) cells in colorectal cancer (CRC) by analyzing tumor gene expression profiles. A total of 1244 CRC patients were involved, and they found that the abundance of mvE cells does not mirror angiogenesis, but rather is associated with the mature blood vessels in the tumor microenvironment and predicts response to chemotherapy as well as patient survival in CRC.
The authors concluded the abundance of mvE cells using xCell algorithm on tumor transcriptome, finding its association with the number of mature blood vessels in a tumor microenvironment and its ability to predict response to chemotherapy, thereby patient survival in CRC.
Although the authors investigated numerous findings, the manuscript was not well summarized and was exceedingly difficult to understand. Too much information and figures significantly decreased readability. Overall, the sample size was large enough, and the findings were novel. The interpretation of the results was reasonable. However, I would like to recommend that the authors revise the manuscript to enhance their study.
1. There were too many results, and I could not understand which one was important. Please show only the important figures you would like to address and put the other information into a supplemental figure or remove it.
2. I could not see any description of the demographical characteristics of the entire cohort, except this study included 1244 CRC patients. For example, I could not see how many samples you investigated among 1244 patients. Did you obtain samples from all patients, or were there any missing samples? I assume that you could not investigate all patients based on the No. at risk in the K-M curve, which were 445 and 149. Please provide detailed information about patient numbers, and I would like to recommend the authors to make a table for better understanding.
3. Did you use resected specimens or biopsy specimens? It seems that the authors used resected specimens, but please clarify.
4. Please use a multivariate analysis if you want to mention mvE is correlated with prognosis.
Round 2
Reviewer 1 Report
Thank you for the revised version of the manuscript. It is well written and all issues were answered regarding the questions that have been made in the first round.
Editor and autors should discuss if maybe some of the figures should been shown in the text or in the supplement as well.